# Metabolomics Analysis Revealed Significant Metabolic Changes in Brain Cancer Cells Treated with Paclitaxel and/or Etoposide

**DOI:** 10.3390/ijms232213940

**Published:** 2022-11-11

**Authors:** Ahlam M. Semreen, Leen Oyoun Alsoud, Waseem El-Huneidi, Munazza Ahmed, Yasser Bustanji, Eman Abu-Gharbieh, Raafat El-Awady, Wafaa S. Ramadan, Mohammad A.Y. Alqudah, Mohd Shara, Ahmad Y. Abuhelwa, Nelson C. Soares, Mohammad H. Semreen, Karem H. Alzoubi

**Affiliations:** 1Department of Pharmacy Practice and Pharmacotherapeutics, College of Pharmacy, University of Sharjah, Sharjah 27272, United Arab Emirates; 2Research Institute for Medical Health Sciences, University of Sharjah, Sharjah 27272, United Arab Emirates; 3Department of Medicinal Chemistry, College of Pharmacy, University of Sharjah, Sharjah 27272, United Arab Emirates; 4Department of Basic Medical Sciences, College of Medicine, University of Sharjah, Sharjah 27272, United Arab Emirates; 5Department of Basic and Clinical Pharmacology, College of Medicine, University of Sharjah, Sharjah 27272, United Arab Emirates; 6School of Pharmacy, The University of Jordan, Amman 11942, Jordan; 7Department of Clinical Sciences, College of Medicine, University of Sharjah, Sharjah 27272, United Arab Emirates; 8Department of Clinical Pharmacy, Faculty of Pharmacy, Jordan University of Science and Technology, Irbid 22110, Jordan

**Keywords:** glioblastoma, untargeted metabolomics, U373, U87, paclitaxel, etoposide, UHPLC-ESI-QTOF-MS, metabolites, metabolic pathways

## Abstract

Cancer of the central nervous system (CNS) is ranked as the 19th most prevalent form of the disease in 2020. This study aims to identify candidate biomarkers and metabolic pathways affected by paclitaxel and etoposide, which serve as potential treatments for glioblastoma, and are linked to the pathogenesis of glioblastoma. We utilized an untargeted metabolomics approach using the highly sensitive ultra-high-performance liquid chromatography-electrospray ionization quadrupole time-of-flight mass spectrometry (UHPLC-ESI-QTOF-MS) for identification. In this study, 92 and 94 metabolites in U87 and U373 cell lines were profiled, respectively. The produced metabolites were then analyzed utilizing *t*-tests, volcano plots, and enrichment analysis modules. Our analysis revealed distinct metabolites to be significantly dysregulated (nutriacholic acid, L-phenylalanine, L-arginine, guanosine, ADP, hypoxanthine, and guanine), and to a lesser extent, mevalonic acid in paclitaxel and/or etoposide treated cells. Furthermore, both urea and citric acid cycles, and metabolism of polyamines and amino acids (aspartate, arginine, and proline) were significantly enriched. These findings can be used to create a map that can be utilized to assess the antitumor effect of paclitaxel and/or etoposide within the studied cancer cells.

## 1. Introduction

The prevalence of central nervous system (CNS) tumors has increased in recent years, especially in adults. As of 2020, they were the 19th most prevalent type of cancer, accounting for 1.7% of all new cancer diagnoses globally [1]. Glioma is the most common form of CNS neoplasm that arises from glial cells, including astrocytes, oligodendrocytes, and ependymal cells. According to the world health organization (WHO), gliomas are graded from I to IV based on the degree of proliferation, aggressiveness, and the presence or absence of necrosis [2]. Sixty-one percent of all primary gliomas appear in the four brain lobes: frontal (25.3%), temporal (19.6%), parietal (12.7%), and occipital (3.3%) [3]. Glioblastoma (GBM), a grade IV brain tumor and the most aggressive type of glioma, had an age-adjusted incidence of 3.2 per 100,000 in the United States [4]. GBMs are almost always found in the brain, but they can also happen in the brain stem, cerebellum, and spinal cord. Adults with GBM have a 90% chance of dying within 24 months of diagnosis [5].

In high-grade glioma, the standard treatment involves surgical resection, concurrent radiation therapy, and temozolomide (TMZ) treatment for 6 weeks, followed by 6 months of adjuvant TMZ therapy [6]. Unfortunately, despite aggressive treatment, including surgery, chemotherapy, and radiotherapy, the median survival time for patients with GBM is only 14.6 months [7]. In addition, GBMs are commonly invasive and located in eloquent areas of the brain, such as those controlling speech, motor function, and senses; therefore, extensive and complete surgical resection is challenging [8].

Metabolomics is an emerging technique that offers unique insights into the pathogenicity of the disease by identifying disease-related determinants and treatment responses [9]. Analysis of the downstream molecular effects of treatment through untargeted MS-based metabolomics approaches could aid in the quest for new candidate biomarkers for treatment response monitoring and novel therapeutic targets. Furthermore, it enables to map within cancer cells of the biochemical pathways targeted by drugs [10]. A previous study explored the metabolic changes in temozolomide-sensitive and temozolomide-resistant GBM cell lines modulated upon cell treatment with temozolomide and lomeguatrib [11].

However, up to date, limited studies have been conducted to investigate the effect of anticancer drugs on the metabolism of GBM. Prior research has found that paclitaxel, a taxoid antineoplastic, might be preferable in metastatic rather than in primary brain cancers [12]. Moreover, etoposide, a podophyllotoxin derivative, crosses the blood–brain barrier and has potential activity against reoccurring malignant gliomas [13]. This is the first study to examine the impacts of anticancer medications paclitaxel and/or etoposide on the metabolic signature of two well-characterized and used brain cancer cell lines (U87 and U373); while utilizing the highly sensitive ultra-high-performance liquid chromatography-electrospray ionization quadrupole time-of-flight mass spectrometry (UHPLC-ESI-QTOF-MS) analytical technology. Investigation of the metabolic effects of paclitaxel and/or etoposide on cancer cells could increase our understanding of treatment response and/or drug resistance associated with these treatments. 

## 2. Results

A total of 32 paclitaxel and/or etoposide-treated cancer cell samples, 4 independent biological replicates from each group (DMSO, paclitaxel 4.2 nM, etoposide 10 µg/mL, and a combination of paclitaxel 4.2 nM and etoposide 10 µg/mL) for each cell line were examined twice by LC-QTOF MS, resulting in 15,000 characteristics metabolites (64 UHPLC-QTOF analyses). After filtration, a total of 92 metabolites for U87 cells treated with (paclitaxel 4.2 nM and/or etoposide 10 µg/mL) and 94 metabolites for U373 treated with (paclitaxel 4.2 nM and/or etoposide 10 µg/mL) were found.

### 2.1. Metabolites in U87 Cell Line

Using one-way ANOVA, we observed that the metabolic profiles of treated U87 and U373 brain cancer cells were vastly different from those of cells treated with DMSO (refer to Appendix A). Student’s *t*-test was used to identify the significantly perturbed metabolites, the fold change was used to assess dysregulated metabolites, and volcano plots were performed to express the significantly altered metabolites concerning fold change. The comparison was as follows: paclitaxel 4.2 nM/DMSO; etoposide 10 µg/mL/DMSO, and paclitaxel 4.2 nM + etoposide µg/mL /DMSO. A total of 26 metabolites were shown to be statistically significant when comparing the control group (DMSO) with the paclitaxel 4.2 nM group (Table 1A). Mainly ADP, guanosine, guanine, hypoxanthine, and nutriacholic acid were the most dysregulated (Figure 1A).

On the other hand, when comparing DMSO with etoposide 10 μg/mL, there were no significant results. Nonetheless, a total of 26 metabolites were significant between DMSO and paclitaxel 4.2 nM + etoposide 10 μg/mL (Table 1B). Mainly ADP, guanine, guanosine, thymidine, mevalonic acid, and nutriacholic acid were the most dysregulated (Figure 1B).

Principal component analysis (PCA) indicated a clear separation between the control group and treatment groups. Figure 2A,B show that cells from every two groups tended to cluster in a concentrated manner without any overlapping, elucidating that there is a difference between the groups. The Venn diagram for the U87 cell line is demonstrated in Figure 2C, showing that 25 metabolites are common between paclitaxel and paclitaxel + etoposide treatments. Shared and uniquely dysregulated metabolites for the U87 cell line were also identified in Table 2. Additionally, one-way ANOVA analysis demonstrated that 33 metabolites in the U87 cells treated with paclitaxel 4.2 nM and/or etoposide 10 μg/mL were statistically significant (Appendix A). The heat map conducted using MetaboAnalyst 5.0 from treated U87 cells with paclitaxel and/or etoposide showed complete separation between DMSO and etoposide; however, paclitaxel and combination treatment showed overlapping (Appendix A).

### 2.2. Enrichment Analysis for U87 Cell Line

The sets of significantly altered metabolites were uploaded to MetaboAnalyst software 5.0 to test for enriched pathways defined using the SMPBD database. The enrichment analysis results are shown in Figure 3A,B which included statistically significant metabolites (*p* < 0.05). Figure 3A,B revealed that purine metabolism, spermidine and spermine biosynthesis, glutathione metabolism, urea cycle, and glycine and serine metabolism were highly enriched in both paclitaxel and combination treatments. However, arginine and proline metabolism were specific to combination treatment (Figure 3B).

### 2.3. Metabolites in U373 Cell Line

Only one metabolite in the U373 cell line was significantly upregulated (adenosine monophosphate) upon treatment with paclitaxel alone (Figure 4A). However, seven metabolites were significantly altered when comparing DMSO with etoposide treatment (Table 3A). Specifically, N-acetylserotonin, diaminopimelic acid, sorbitol, L-arginine, adenosine monophosphate, deoxyguanosine, and L-phenylalanine were the most dysregulated (Figure 4B). Interestingly, eighteen metabolites were shown to be significantly affected when comparing DMSO with the combination of etoposide 10 μg/mL and paclitaxel 4.2 nM (Table 3B). L-arginine, guanosine monophosphate, succinylacetone, adenine, diaminopimelic, L-glutamic acid, N-acetylserotonin, and sorbitol were highly dysregulated (Figure 4C). 

The PCA for DMSO with paclitaxel 4.2 nM, DMSO with etoposide 10 μg/mL, and DMSO with (paclitaxel 4.2 nM + etoposide 10 μg/mL) in U373 cell line are shown in Figure 5. There was an overlap when comparing DMSO with paclitaxel (Figure 5A); however, separate clusters were noticed when comparing etoposide and the combination of paclitaxel and etoposide with DMSO (Figure 5B,C). Venn diagram comparing drug metabolites response to drug treatment identified in U373 cell line treated with paclitaxel 4.2 nM and/or etoposide 10 μg/mL showed that only one metabolite is common among the three groups and six metabolites are common between etoposide and the combination groups. These metabolites are listed in Table 4 and demonstrated that adenosine monophosphate was the only common metabolite among all groups. However, none of the metabolites were common between etoposide and combination treatment. In addition, ANOVA for the U373 cell line in Appendix A revealed 22 metabolites to be statistically significant. The heat map from treated U373 cells with paclitaxel and/or etoposide showed complete separation between all the groups (Appendix A).

### 2.4. Enrichment Analysis for U373 Cell Line

In the U373 cell line enrichment analysis: Thiamine, phenylacetate, alanine, and butyrate metabolisms were significantly enriched when treated with paclitaxel 4.2 nM (Figure 6A). However, amino acids metabolism and urea cycle were the most altered metabolic pathways when treated with etoposide 10 μg/mL (Figure 6B). For the combined treatment (paclitaxel 4.2 nM and etoposide 10 μg/mL), purine, phenylalanine and tyrosine, and aspartate metabolism in addition to urea cycle were the most significantly perturbed pathways (Figure 6C).

## 3. Discussion

With the rapid advancement of mass spectrometry technologies and MS-based omics methodologies, metabolomics has been utilized in numerous research fields, such as cancer research and various other diseases. At the molecular level, metabolomics uses novel biomarkers to investigate disease origin [14]. This is the first to analyze metabolites and metabolic pathways of U373 and U87 cancer cell lines treated with paclitaxel (4.2 nM) and/or etoposide (10 μg/mL).

The constructed volcano plots expressed dysregulated metabolites related to GBM. Nutriacholic acid was found to be increased also, mevalonic acid (mevalonate) were found to be slightly increased in the U87 cell line upon treatment with paclitaxel, both alone and in combination with etoposide (see Figure 1A,B). Nutriacholic acid is a well-known bile acid (natural detergent that helps the intestine and liver solubilize fats and sterols for absorption or excretion) [15]. Lu et al. reported that nutriacholic acid was associated with hepatocarcinoma and tumor mutagenesis [16]. Mevalonate metabolism provides cancer and immune cells with various products that ensure cell functionality. Many studies have found upregulation of the mevalonate pathway in a wide range of cancers, including leukemia, lymphoma, multiple myeloma, breast, hepatic, pancreatic, esophageal, and prostate cancers [17].

In our study, ADP was found to be decreased in the U87 cell line upon treatment with paclitaxel alone and in combination with etoposide. Researchers have shown that activated platelets promote cancer cell proliferation and tumor formation by secreting adenosine diphosphate (ADP), a primary mediator of tumor cell-induced platelet aggregation (TCIPA) [18].

Furthermore, guanosine was observed to be decreased in the U87 cell line upon treatment with paclitaxel alone and in combination with etoposide. In fact, the potential effect of guanosine reported by Belluardo and his colleagues indicates that guanosine promotes neuroblastoma cell differentiation, reducing the chance of cancer spread [19].

The volcano plot for U373 cells treated with paclitaxel and etoposide (see Figure 4C) demonstrated that N-acetyl-L-alanine was dysregulated. N-Acetyl-L-alanine belongs to N-acyl-alpha amino acids organic compounds [20]. Interestingly, N-Acetyl-L-Aspartic acid (NAA) also belongs to the N-acyl-alpha amino acids compounds pathway and is highly involved in tumor growth; therefore, it is considered a target for anticancer treatment [21]. Thus, the observed increase of these tumor-associated metabolites upon treatment suggests this is a possible mechanism for cancer growth persistence despite treatment with anticancer agents.

In this study, arginine was increased in the two groups, DMSO with etoposide and DMSO, and combination in the U373 cell line. The complex role of L-arginine in immunomodulation includes possibly boosting antitumor immunity in some situations. Lack of arginine has been shown to increase immune response and induce cancer cell death independently [22].

Functional enrichment analysis showed that drug treatment with paclitaxel 4.2 nM in the U87 cancer cell line fundamentally affected amino acid metabolism, including that of glycine and serine, as well as the metabolism of purines, pyrimidines, and glutathione. Furthermore, when treated with paclitaxel, urea and citric acid cycles and polyamine metabolism were significantly enriched. Interestingly, treatment with paclitaxel 4.2 nM + etoposide 10 μg/mL revealed similar enrichment results as seen with treatment with paclitaxel alone, alteration of arginine and proline metabolism were found to be more significantly associated with the combination treatment. It has been reported that impaired purine metabolism is linked to cancer development because purines are essential nucleotide building blocks in cell growth [23]. Particularly, purine metabolism was shown to be deregulated in hepatocellular carcinoma [24]. Purine and pyrimidine molecules are synthesized during nucleotide metabolism, which is crucial for DNA replication, RNA synthesis, and cell energy production. In fact, cancer is characterized by the uncontrolled growth of tumors owing to increased nucleotide metabolism [25].

On the other hand, the observed dysregulation in the urea cycle could be attributed to the fact that cancers are characterized by urea cycle dysregulation (UCD); basically, through increasing nitrogen utilization for pyrimidine synthesis. UCD produces nucleotide imbalances that can be detected in cancer patients’ samples based on mutation patterns and biochemical signatures. Immunotherapy is more effective in patients with UCD, but the prognosis is worse [26]. Nitrogen can be utilized to synthesize urea cycle intermediates outside of the liver through the differential expression of urea cycle enzymes based on cellular needs. It has been noted that urea cycle enzymes are expressed differently in cancer, revealing a unique mechanism by which nitrogen is incorporated into biomass more rapidly [27].

Glutathione (GSH), one of the best-known antioxidant tripeptides, is essential in maintaining normal cellular functions such as proliferation, differentiation, and apoptosis [28]. Protection from reactive oxygen species (ROS) damage is achieved by GSH, an antioxidant agent [29]. High ROS and consequent disruptions in GSH homeostasis are hallmarks of cancer cells [30,31]. GSH is an integral part of the brain’s antioxidant defense system. Together with GSH-related enzymes, it protects cells from damage caused by free radicals and controls how tumor cells respond to treatments such as irradiation and chemotherapy [32].

Moreover, the citric acid cycle was significantly impacted in U87 cells treated with paclitaxel alone and in combination with etoposide. The citric acid cycle is one of the essential sources of cellular energy that provides electron carriers with oxidative phosphorylation leading to ATP production through the electron transport chain. In cancer cells, fatty acids are the third fuel source, producing acetyl-CoA through β-oxidation, which enters the citric acid cycle for further oxidation [33]. It is worth mentioning that multiple types of cancer carry mutations disrupt the citric acid cycle. This leads to an imbalance in the citric acid cycle’s metabolite production and is likely a factor in cancer development. In addition, the intermediates of the citric acid cycle can affect carcinogenesis and metastasis [27]. The citric acid cycle metabolic pathway is indispensable for providing cellar energy [34].

In the U373 cell line enrichment analysis, the following metabolites were significantly enriched when treated with paclitaxel 4.2 nM: thiamine, phenylacetate, alanine, and butyrate. In fact, several metabolic processes rely on thiamine, an essential cofactor for numerous critical enzymes [35]. Lu’o’ng et al. reported that high rates of tumor cell survival, growth, and resistance to treatment were associated with thiamine supplementation [36]. Additionally, when given at sufficient doses, thiamine supplementation can stimulate tumor proliferation [37].

Moreover, phenylacetate (PA), an aromatic fatty acid metabolite of phenylalanine, has been shown to have potential anticancer action [38]. Franco et al. reported that PA treatment, previously shown to inhibit prostate cancer growth, resulted in renal cancer growth inhibition at doses of 2–5 mM and an increase in cells in G1 after 24 h and was previously shown to inhibit prostate cancer growth [39]. These previous findings might explain its association with GBM.

Likewise, butyrate, a short-chain fatty acid, is well-known for its potential as a secondary chemopreventive as it was perceived to diminish colon cancer cell growth and stimulate apoptosis [40]. In the colon, butyrate is produced by beneficial, commensal bacteria and has been shown to have remarkable anticancer effects. In particular, butyrate has a skewed inhibitory impact on the cell development of malignant colonocytes while serving as the primary energy source for normal colonocytes [41].

In the current study, we found that aspartate metabolism was significantly altered in U373 cells treated with monotherapy of either paclitaxel or etoposide or a combination. Researchers have shown that aspartate affects neurological and neuroendocrine signaling with implications for human health over the past two decades [42]. Therefore, aspartate may limit tumor growth, and its availability could be targeted for cancer treatment. Due to the outgrowth of solid tumors, cancer cells live in nutrient and oxygen-deficient conditions [43]. Upon receiving an action potential from the presynaptic terminal, aspartic acid is released across the synaptic cleft and binds with specific receptors on the postsynaptic membrane [44].

Using functional enrichment analysis, we found that paclitaxel and etoposide therapies significantly impacted production pathways critical for regulating energy production and which have previously been connected to cancer initiation, progression, and aggressiveness. Cancer cells rely on amino acids for proliferation because of protein anabolism, and it is well-known that cancer cells have poor amino acid metabolism. Aside from their probable function in ATP production, amino acids are also required for cellular redox homeostasis and nucleoside synthesis, both known to be aberrant in cancer [45,46].

## 4. Materials and Methods

### 4.1. Reagents and Materials

Formic acid was purchased from Fisher Chemical (Loughborough, UK). Methanol, acetonitrile, deionized water, and LC-MS CHROMASOLV were obtained from Honeywell (Wunstorfer Strasse, Seelze, Germany). U87 and U373 cells were purchased from the Radiobiology and Experimental Radio Oncology Lab, University Cancer Center Hamburg, Hamburg, Germany. Etoposide was purchased from Sigma-Aldrich (St. Louis, MO, USA). Paclitaxel was purchased from Merck (Darmstadt, Germany). Fetal bovine serum, penicillin, and streptomycin were purchased from Sigma Aldrich (St. Louis, MO, USA).

### 4.2. Cell Culture

The two brain cancer cell lines, U87 and U373, were grown as monolayers in Dulbecco’s modified eagle medium (DMEM) media with 10% fetal bovine serum and 1% penicillin/streptomycin. All the cell cultures were incubated at 37 °C in a humidified environment with 5% CO_2_. The culture medium was changed every 2–3 days. Precisely, these two cell lines have not been previously studied while utilizing paclitaxel and etoposide.

### 4.3. Treatment of Cells with Anticancer Drugs

Two million cells (per type) were seeded in a 75 cm^2^ tissue culture flask and incubated overnight. Dimethyl sulfoxide (DMSO) (0.5%) was added to the control cells for 24 h. U87 cells were treated with paclitaxel 4.2 nM and/or etoposide 10 µg/mL for 24 h. U373 cells were treated with paclitaxel 4.2 nM and/or etoposide 10 µg/mL for 24 h. Cells were obtained as pellets by trypsinization after incubation, and they were then twice-washed with phosphate-buffered saline (PBS) before being resuspended in 1 mL PBS for analysis. After another round of centrifugation at 1200 rpm for 10 min at room temperature, cells were again collected as pellets. Three duplicate flasks for each treated cell line were created for every analysis. During the incubation process, cells were maintained under the same conditions. Cell collection was carried out simultaneously for all the samples to eliminate the influence of circadian rhythms’ influence on the cells’ response to treatments.

### 4.4. Cells Samples Preparation and Metabolites Extraction

Each flask contained two million cells to eliminate the potential for variation through differing cell densities. Prior to extracting the cells, 1 mL of the extraction solvent (methanol + 0.1% formic acid) was added to each microcentrifuge tube containing cells. This step effectively halted metabolic activity in the cells. After being put on ice for an hour, the cells were vortexed for two minutes to maximize metabolite extraction.

The insoluble cell matrices were ultrasonically disrupted using a QSONICA sonicator (Qsonica, Newtown, CT, USA) at 30% amplifier power for 30 s in a cold bath. After centrifuging the cell debris (15,000 rpm for 10 min at 24 °C) to separate the cell wall from the other cellular components, the supernatants containing the cellular metabolites were collected and transferred to liquid chromatography (LC) glass vials to evaporate the solvent using an EZ-2 Plus (GeneVac, Ipswich, UK). We pooled a similar volume of 10 μL from each sample to prepare quality control samples and analyzed them through UHPLC-QTOF-MS. After being dried, samples containing the necessary metabolites were resuspended in 200 µL (water + 0.1 percent formic acid) and vortexed for 2 min to ensure a uniform mixture. In order to prepare the samples for QTOF MS analysis, they were filtered through a 0.45 µm pore size hydrophilic nylon syringe filter and then added to the inserts of LC glass vials.

### 4.5. Tandem Liquid Chromatography Mass Spectrometry (QTOF MS) Conditions

A QTOF MS and Elute UHPLC were utilized to separate and detect the metabolites in the cells (Bruker, Bremen, Germany). Thermostatically controlled column compartment, autosampler (Elute UHPLC), and solvent delivery pump (HPG 1300) constituted the system. Bruker Compass HyStar 5.0 SR1 Patch1 (5.0.37.1), Bruker Compass 4.1 for otofSeries, and otofControl Version 6.0 were all used as the data management software. Microsoft Windows 10 Enterprise 2016 Long Term Servicing Branch was the operating system used.

The mobile phase gradient scheme is shown in Table 5. A 10 µL sample was injected, and the separation was performed in a column oven at 35 °C using Hamilton^®^ Intensity Solo 2 C18 column (100 mm 2.1 mm 1.8 m).

There were two acquisition segments; auto MS scan with sodium formate, which ranged from 0 to 0.3 min, and auto MS/MS with fragmentation, which ranged from 0.3 to 30 min. In both segments, the acquisition was implemented in the positive mode at a frequency of 12 Hz. Metabolites were analyzed using electrospray ionization (ESI) in the 20–1300 *m/z* range. The ESI source with dry nitrogen gas had a flow rate of 10 L/minute and a drying temperature of 220 °C. The ESI’s capillary voltage was 4500 V at 2.2 bar nebulizer pressure. For MS2 acquisition, collision energy was set to 20 eV and end plate offset to 500 V.

To test the column’s and the mass spectrometer’s performances, a test mixture of (TRX-2101/RT-28-calibrants for Bruker T-ReX LC-QTOF solution) was used. Additionally, the performance of reversed-phase liquid chromatography (RPLC) separation and multipoint retention time calibration were tested using (TRX-3112-R/MS certified human serum for Bruker T-ReX LC-QTOF solution from Nova Medical Testing Inc., Edmonton, AB, Canada).

### 4.6. Data Processing and Analysis

MetaboScape^®^ 4.0 software was used for data processing and statistical analysis (Bruker, Bremen, Germany) [47]. These settings were defined for molecular feature detection in the T-ReX 2D/3D workflow: a minimum intensity threshold of 1000 counts, a minimum peak duration of 7 spectra for peak detection, and a peak area for feature quantification during bucketing. The mass recalibration was completed within a retention time range of 0 to 0.3 min. Only features found in at least four of the sixteen samples (per cell type) were considered. The MS/MS import method, on the other hand, was set to be done on average. The following data bucketing parameters were assigned: The retention duration ranged from 0.3 to 25 min and the mass range extended from 50 to 1000 *m*/*z*.

The robustness of the MetaboScape program is exhibited through the generation of bucket statistics and box plots displaying statistics for each metabolite displayed by the compound ID of the selected bucket across all analyses included in the current experiment (14). With bucket statistics, it is possible to quickly compare metabolite intensities among groups. The box plot, on the other hand, makes it easy to spot differences between groups by graphically depicting the middle and outer quartiles of metabolite levels. In order to ensure proper identification, the following criteria were established: MS/MS spectra and retention time (RT) were used to characterize the chemicals that were initially unknown from QTOF MS data. Compounds that passed the screening and displayed MS/MS or MS/MS in conjunction with RT were annotated with the help of the Human Metabolome Database (HMDB) 4.0, a database of annotated metabolomics resources, spectrum library [48]. All the compounds that were ultimately selected were compared to this library. The collision-induced dissociation (CID) information can be found in Appendix A.

The HMDB 4.0 was used for mapping MS/MS spectra and retention times to identify metabolites. When more than one feature matched a given database entry, these metabolites were then filtered by using the entry of each metabolite with the highest annotation quality score (AQ score) among several entries related to the same metabolite, i.e., the best fit with the most factors including, retention time, MS/MS, *m/z* values, analyte list, msigma, and spectral library. Using the abovementioned factors, we chose only one of the repeated metabolites with the same ID and name but different *p*-value.

As part of the analysis, metabolite data were exported as CSV files and imported into MetaboAnalyst 5.0 software (https://www.metaboanalyst.ca, accessed on 20 April 2022), a comprehensive metabolomics platform [49]. Two-tailed independent student *t*-tests were used to distinguish significantly altered metabolites for each drug compared to DMSO. A volcano plot depicting statistical significance and fold change for cellular metabolite dysregulation was constructed for each condition. Multiple groups were compared using a one-way analysis of variance (ANOVA).

The significance level was set at *p* < 0.05. The principal component analysis (PCA) was also carried out using MetaboAnalyst 5.0 software to compare the two groups. Multiple hypothesis testing was corrected, and false positives were reduced using the false discovery rate (FDR). Venn diagrams were assembled using (http://bioinformatics.psb.ugent.be/webtools/Venn/, accessed on 24 July 2022) in order to compare the overlap of dysregulated metabolites from each treatment comparison group.

## 5. Conclusions

The present results demonstrated that treatment with paclitaxel and/or etoposide significantly altered the major metabolic pathways of the analyzed cell lines (U87 and U373). These findings have important implications for translational cancer research as they may lead to the identification of novel candidate biomarkers for monitoring treatment response and therapy progression in clinical settings among glioblastoma patients, allowing for the development of more efficient and tailored anticancer therapies. The present analysis revealed distinct metabolites to be significantly dysregulated (nutriacholic acid, L-phenylalanine, L-arginine guanosine, ADP, hypoxanthine, and guanine), and to a lesser extent, mevalonic acid. Furthermore, the urea and citric acid cycles, and polyamine and amino acid metabolism were significantly enriched. These newly identified metabolic effects may serve to apprise new potential therapeutic targets and merit further confirmatory studies.

## Figures and Tables

**Figure 1 ijms-23-13940-f001:**
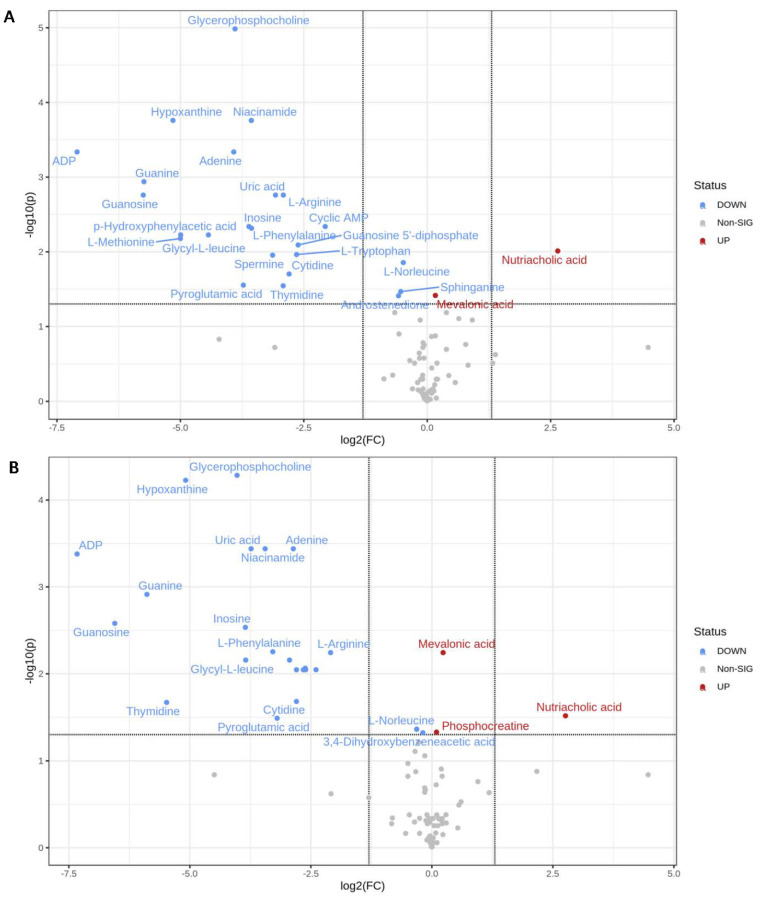
Volcano plots of U87 cells treated with (**A**) DMSO with paclitaxel 4.2 nM; (**B**) DMSO with (paclitaxel 4.2 nM and etoposide 10 μg/mL).

**Figure 2 ijms-23-13940-f002:**
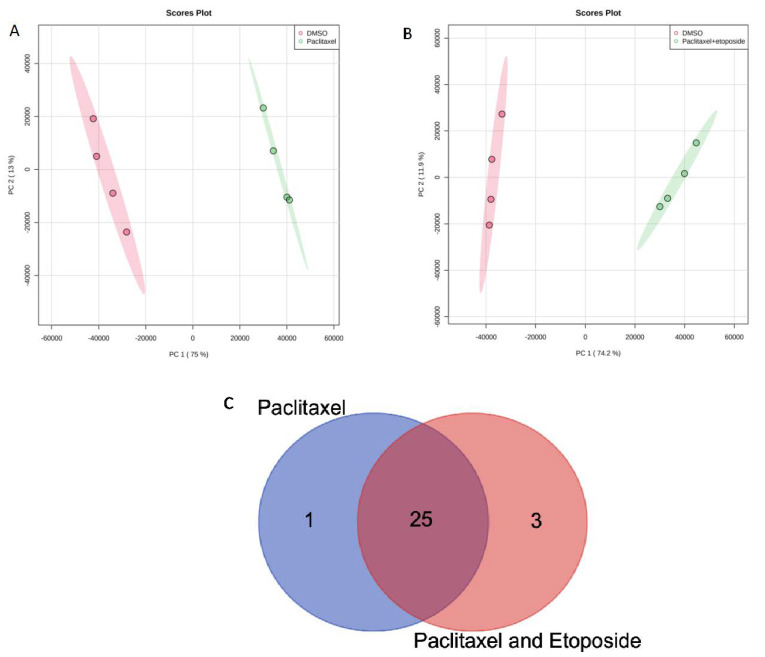
Principal component analysis (PCA) for (**A**) DMSO with paclitaxel 4.2 nM; (**B**) DMSO with (paclitaxel 4.2 nM + etoposide 10 μg/mL); (**C**) Venn diagram comparing metabolites’ response to treatment with paclitaxel 4.2 nM or paclitaxel 4.2 nM + etoposide 10 μg/mL in U87 cell line.

**Figure 3 ijms-23-13940-f003:**
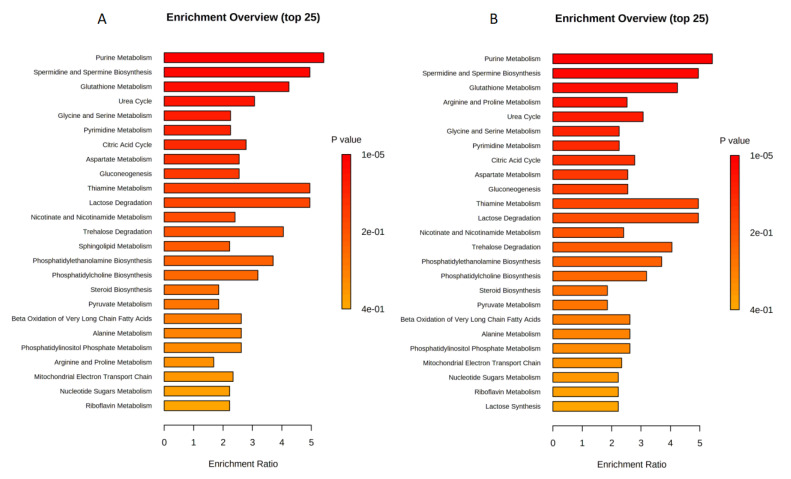
Enrichment analysis for (**A**) DMSO with paclitaxel 4.2 nM; (**B**) DMSO with (paclitaxel 4.2 nM + etoposide 10 μg/mL) in U87 cell line.

**Figure 4 ijms-23-13940-f004:**
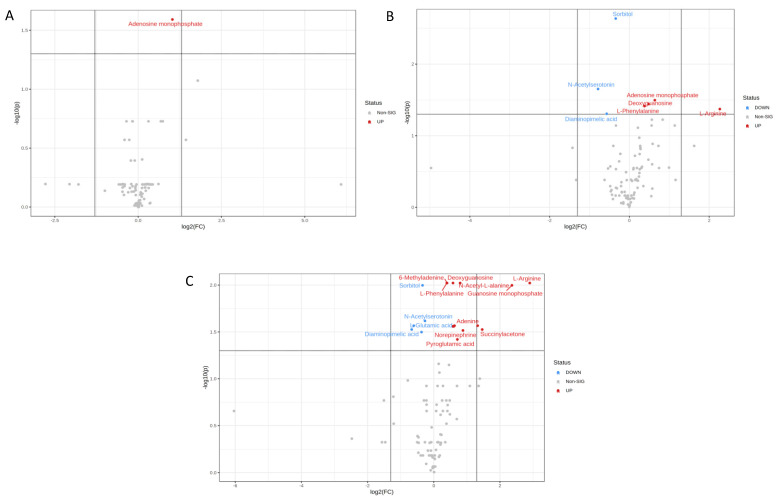
Volcano plots of U373 cells treated with (**A**) paclitaxel 4.2 nM and DMSO; (**B**) DMSO with etoposide 10 μg/mL; (**C**) DMSO with (paclitaxel 4.2 nM + etoposide 10 μg/mL).

**Figure 5 ijms-23-13940-f005:**
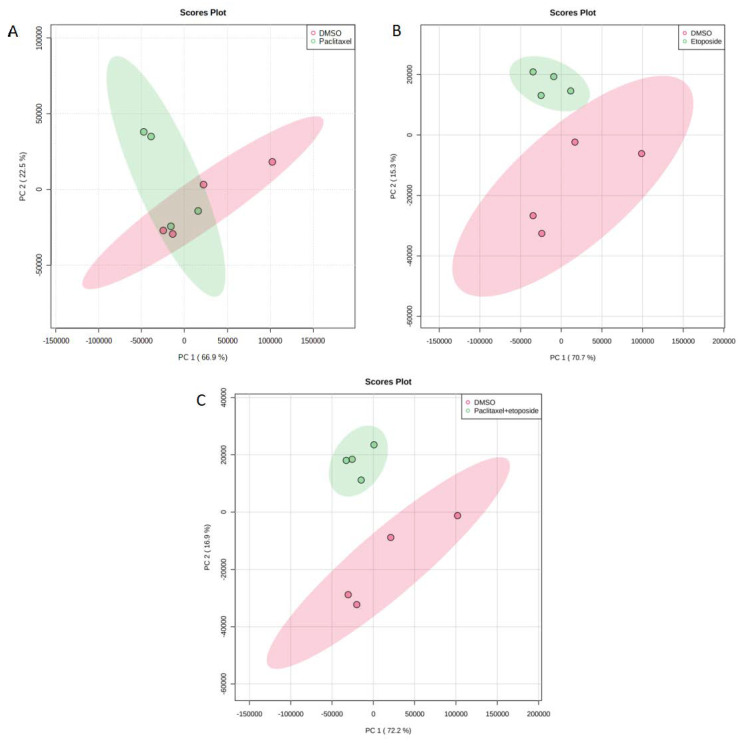
(**A**) PCA for DMSO with paclitaxel 4.2 nM; (**B**) PCA for DMSO with etoposide 10 μg/mL; (**C**) PCA for DMSO with (paclitaxel 4.2 nM + etoposide 10 μg/mL) in U373 cell line.

**Figure 6 ijms-23-13940-f006:**
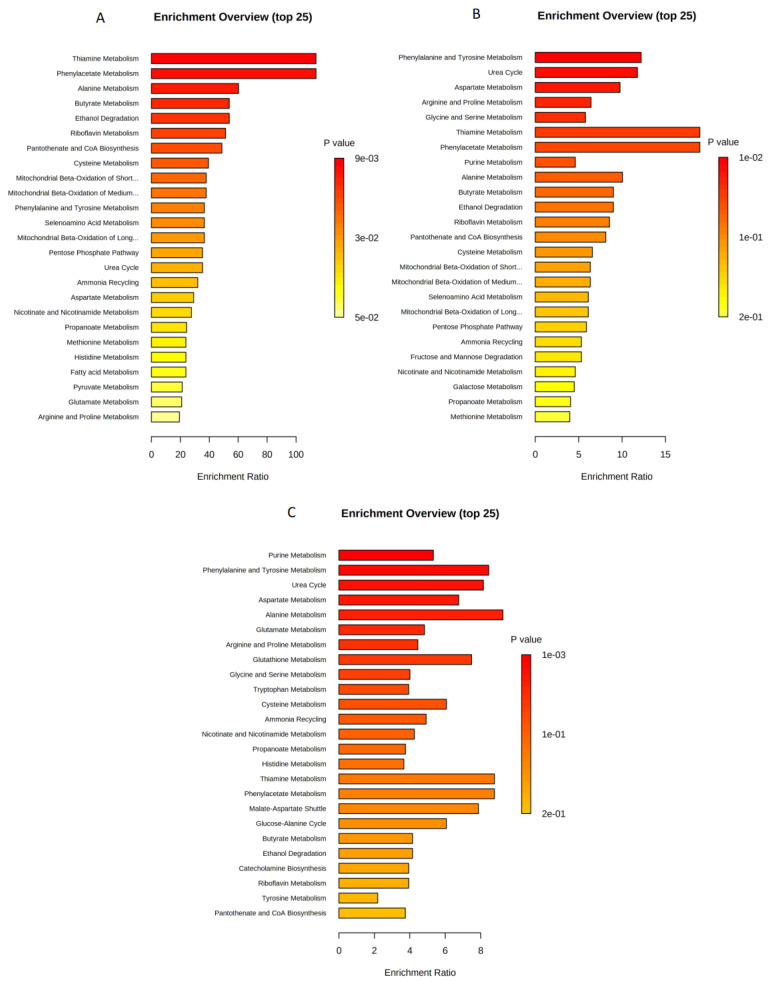
Enrichment analysis for (**A**) DMSO with paclitaxel 4.2 nM; (**B**) DMSO with etoposide 10 μg/mL; (**C**) DMSO with (paclitaxel 4.2 nM + etoposide 10 μg/mL) in U373 cell line.

**Table 1 ijms-23-13940-t001:** Statistically significant metabolites in U87 cells treated with (**A**) paclitaxel 4.2 nM and (**B**) paclitaxel 4.2 nM + etoposide 10 μg/mL.

**A. Statistically Significant Metabolites in U87 Cell Line Treated with Paclitaxel 4.2 nM/mL**				
Metabolite	*t*-stat	*p*-value	FDR	Fold change
ADP	39.479	0.000025	0.00045982	0.0073262
Guanosine	19.744	0.00017013	0.0017391	0.01855
Guanine	27.592	0.000075	0.0011507	0.018726
Hypoxanthine	29.998	0.00000482	0.00017402	0.028151
L-Methionine	11.449	0.0010822	0.0066372	0.031249
p-Hydroxyphenylacetic acid	12.337	0.00090239	0.00593	0.031378
Glycyl-L-leucine	11.288	0.00087297	0.00593	0.046307
Adenine	15.341	0.0000249	0.00045982	0.066085
Glycerophosphocholine	28.954	0.000000113	0.0000104	0.067262
Pyroglutamic acid	6.3228	0.0066699	0.027892	0.075638
Inosine	13.845	0.0005007	0.00458	0.081824
Niacinamide	19.588	0.00000567	0.00017402	0.084552
L-Phenylalanine	14.813	0.00063069	0.0048353	0.08486
Spermine	7.2734	0.0022883	0.01108	0.11408
Uric acid	22.543	0.00014664	0.0017391	0.11878
Thymidine	6.5862	0.0071226	0.02849	0.13229
L-Arginine	13.13	0.00015563	0.0017391	0.1326
Cytidine	7.7175	0.0045226	0.019813	0.14374
L-Tryptophan	9.8826	0.0021262	0.010867	0.1597
Guanosine 5’-diphosphate	11.515	0.0014062	0.0080855	0.16321
Cyclic AMP	8.3747	0.00054761	0.00458	0.23873
Androstenedione	4.5213	0.010954	0.038761	0.66828
Sphinganine	4.9041	0.0088823	0.034049	0.68976
L-Norleucine	6.6297	0.0030279	0.013928	0.71434
Mevalonic acid	−3.8293	0.010428	0.038375	1.1212
Nutriacholic acid	−10.591	0.0017986	0.0097337	6.25
**B. Statistically significant metabolites in U87 cell line treated with paclitaxel 4.2 nM /mL + etoposide 10 μg/mL**				
Metabolite	*t*-stat	*p*-value	FDR	Fold change
ADP	39.641	0.0000273	0.00041814	0.0062162
Guanosine	20.192	0.00022842	0.0026268	0.010662
Guanine	27.875	0.0000928	0.0012194	0.016895
Thymidine	7.3937	0.0048716	0.021342	0.022396
Hypoxanthine	28.231	0.00000129	0.0000592	0.029428
Glycerophosphocholine	35.644	0.000000565	0.000052	0.061413
Inosine	13.625	0.00028557	0.0029192	0.069193
Glycyl-L-leucine	11.044	0.0009866	0.0069563	0.069508
Uric acid	21.477	0.0000139	0.00036381	0.075097
Niacinamide	20.815	0.000019	0.00036381	0.091824
L-Phenylalanine	14.45	0.00060549	0.0055705	0.10242
Pyroglutamic acid	6.1554	0.0081168	0.032467	0.10904
Cyclic AMP	10.615	0.0010586	0.0069563	0.13018
Adenine	13.059	0.0000198	0.00036381	0.13747
Cytidine	7.7175	0.0045226	0.020804	0.14374
L-Methionine	10.229	0.0018547	0.0089805	0.14382
p-Hydroxyphenylacetic acid	10.849	0.0016757	0.0089805	0.15864
Guanosine 5’-diphosphate	11.515	0.0014062	0.0086245	0.16321
L-Tryptophan	9.6991	0.0017879	0.0089805	0.16393
Spermine	5.7422	0.0016656	0.0089805	0.1902
L-Arginine	7.0817	0.00070717	0.0057029	0.23436
L-Norleucine	3.6225	0.0113	0.043318	0.80446
3,4-Dihydroxybenzeneacetic acid	3.4628	0.013422	0.047493	0.93086
Phosphocreatine	−3.5105	0.012692	0.046708	1.0684
Mevalonic acid	−6.3456	0.00074386	0.0057029	1.1743
Nutriacholic acid	−6.5371	0.0072758	0.030426	6.7728

FDR: false discovery rate.

**Table 2 ijms-23-13940-t002:** Metabolites that responded to drug treatments according to Venn diagram comparison in U87 cell line.

Groups	Metabolites
Common metabolites between paclitaxel and paclitaxel + etoposide treatments (25)	Niacinamide Nutriacholic acid Glycyl-L-leucine L-Arginine L-Norleucine Guanosine 5’-diphosphate Guanine Cytidine Thymidine Sphinganine Spermine p-Hydroxyphenylacetic acid L-Tryptophan Cyclic AMP Mevalonic acid Glycerophosphocholine ADP Hypoxanthine L-Methionine Pyroglutamic acid Adenine Guanosine L-Phenylalanine Uric acid Inosine
Unique metabolite from paclitaxel treatment (1)	Androstenedione
Unique metabolites from paclitaxel and etoposide treatment (3)	3,4-Dihydroxybenzeneacetic acid Saccharopine Phosphocreatine

**Table 3 ijms-23-13940-t003:** Statistically significant metabolites in U373 cells treated with paclitaxel 4.2 nM and/or etoposide 10 μg/mL.

**A. Statistically Significant Metabolites in U373 Cells Treated with Etoposide 10 μg/mL**				
	*t*-stat	*p*-value	FDR	Fold Change
N-Acetylserotonin	9.5789	0.00047303	0.022232	0.58014
Diaminopimelic acid	4.6918	0.003674	0.049336	0.67382
Sorbitol	11.637	0.0000246	0.002309	0.78847
L-Phenylalanine	−5.9766	0.0020496	0.038532	1.3019
Deoxyguanosine	−5.4895	0.0015345	0.036061	1.397
Adenosine monophosphate	−6.3756	0.0010177	0.031888	1.5557
L-Arginine	−5.0157	0.0027084	0.042432	4.8143
**B. Statistically significant metabolites in U373 cells treated with paclitaxel nM /mL + etoposide 10 μg/mL**				
	*t*-stat	*p*-value	FDR	Fold Change
Diaminopimelic acid	4.4706	0.0045103	0.029796	0.62996
L-Glutamic acid	5.0364	0.0031745	0.027128	0.65732
3a,6b,7b-Trihydroxy-5b-cholanoic acid	5.0002	0.0057423	0.031751	0.77372
Sorbitol	7.9213	0.00064331	0.01006	0.79015
N-Acetylserotonin	8.7249	0.0020526	0.024118	0.83152
L-Phenylalanine	−10.2	0.00012294	0.0095052	1.3149
6-Methyladenine	−7.3836	0.0005056	0.0095052	1.32
L-Tryptophan	−4.6148	0.0038147	0.027583	1.4871
Deoxyguanosine	−6.8588	0.00047465	0.0095052	1.4964
Glycerophosphocholine	-4.9201	0.0036767	0.027583	1.5182
Adenine	-4.8976	0.0028844	0.027113	1.5443
Pyroglutamic acid	−4.333	0.0072763	0.037998	1.6388
N-Acetyl-L-alanine	−8.0485	0.00027672	0.0095052	1.7372
Norepinephrine	−5.415	0.0051875	0.030476	1.8433
Adenosine monophosphate	−7.0313	0.0028079	0.027113	2.5075
Succinylacetone	−7.2032	0.0047548	0.029796	2.7582
Guanosine monophosphate	−6.8338	0.00074918	0.01006	5.1413
L-Arginine	−7.9259	0.000333	0.0095052	7.4674

**Table 4 ijms-23-13940-t004:** Metabolites that responded to drug treatments according to Venn diagram comparison in U373 cell line.

Groups	Metabolites
Common metabolites between paclitaxel and paclitaxel + etoposide treatments (1)	Adenosine monophosphate
Unique metabolites from paclitaxel and etoposide treatment (6)	N-Acetylserotonin L-Arginine Diaminopimelic acid Sorbitol Deoxyguanosine L-Phenylalanine
Unique metabolites from paclitaxel and etoposide treatment (11)	6-Methyladenine Succinylacetone Guanosine monophosphate L-Glutamic acid L-Tryptophan N-Acetyl-L-alanine Glycerophosphocholine Pyroglutamic acid Adenine Norepinephrine 3a,6b,7b-Trihydroxy-5b-cholanoic acid

**Table 5 ijms-23-13940-t005:** The mobile phase gradient scheme.

Time Point (min)	Mobile Phase Concentration (A = Water with 0.1% Formic Acid; B = Acetonitrile with 0.1% Formic Acid)	Flow Rate
0–2	99% A and 1% B	0.25 mL/min
2–17	99 to 1% A and 1 to 99% B	0.25 mL/min
17–20	99% B and 1% A	0.25 mL/min
20–20.1	99% B changed to 99% A	0.35 mL/min
20.1–28.5	99% A changed to 1% B	0.35 mL/min
28.5–30	99% A changed to 1% B	0.25 mL/min

## Data Availability

Data is contained within the article and Supplementary Material.

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
