# Peer review of "Metabolomics Analysis Revealed Significant Metabolic Changes in Brain Cancer Cells Treated with Paclitaxel and/or Etoposide"

_ijms, 2022, doi:10.3390/ijms232213940_

Round 1

Reviewer 1 Report

General Comments

Reviewed is the manuscript “Metabolomics Analysis Revealed Significant Metabolic Changes in Brain Cancer Cells Treated with Paclitaxel and/or Etoposide” submitted by Selina Schelbert, et, al.

The authors found that various metabolites, including nutriacholic acid, mevalonic acid, L-phenylalanine, L-arginine, guanosine, ADP, hypoxanthine, and guanine, were shown to be highly dysregulated in paclitaxel and/or etoposide-treated cells.

The metabolism of polyamines and amino acids (aspartate, arginine, and proline) as well as the urea and citric acid cycles were also greatly enriched.

The description of each strategy is well-organized and the content flows smoothly throughout.

Although the article is well-written, has few spelling errors, and is presented in a highly professional manner in terms of style and layout, there are issues with the presentation and analysis as it is now written.

The material will still meet the requirements for publication after minimal changes.

Those comments are from a statistical perspective:

·         In line 107, the authors stated that mevalonic acid was one of the most dysregulated metabolites. However, the fold change in table 1 for mevalonic is only 1.1212, will this one just be a false positive?

·         Similarly, in figure 1B, phosphocreatine is one of the most dysregulated ones, but the fold change is low (1.0684). Should be more careful when drawing conclusions like that.

·         Similar problem when drawing conclusions in figure 4, both fold change and FDR should be used as selecting criteria.

·         The authors mentioned that “MS/MS spectra and retention time (RT) were used to characterize the chemicals that were initially unknown from QTOF MS data”. It is highly recommended that CID information should be provided in the supplementary files.

Reviewer 2 Report

Through the research article “Metabolomics Analysis Revealed Significant Metabolic Changes in Brain Cancer Cells Treated with Paclitaxel and/or Etoposide”, authors Semreen et al have investigated metabolic pathway glioblastoma. They have specifically focused on the identification of metabolites and metabolic pathways that get influenced by

paclitaxel and etoposide treatment in glioblastoma.  This study highlights the relevance of metabolic features for potential treatments advancements.

This study has followed a simple and clear experimental strategy and the two cell lines used are investigated for the first time for the impact of Paclitaxel and/or Etoposide. The paper can be accepted in its current form.

Reviewer 3 Report

Dear Authors,

The work, I received for review, presents the results of the studies conducted for the determination of the effect of two anticancer drugs (paclitaxel and etoposide) on the metabolic pathways in glioblastoma. Two well-characterized brain cancer cell lines (U87 and U373) were used as model systems. However, there is no more precise rationale for selecting these two cancer cell lines. You showed that distinct metabolites were significantly dysregulated. This can have important implications for the identification of novel biomarkers / therapeutic targets and the development of more efficient glioblastoma therapies. Overall, your manuscript is complete and nicely written. It contains a large amount of data presented in an orderly and logical manner. Their analysis, based on the current literature references to the topic (many works from 2022), allowed you to draw constructive conclusions. The layout of the work is well thought out and its editing is careful. I noted only some minor errors that should be corrected prior to publication. These are:

1. In Abstract, lines 26-30: This sentence is too long and therefore not legible. I propose to rewrite them and emphasize the aim of the work.

2. Antitumor (line 37) or anti-tumor (line 297)? Chose one form. The same for anticancer / anti-cancer, Q-TOF / QTOF, paclitaxel / Paclitaxel, etoposide / Etoposide, etc.

3. Pay attention to the font at lines 107, 156-157.

4. Let Figure 2C be smaller, proportional to the others.

5. Lines 182-183: Correct this sentence.

6. Lines 438-444: I propose to present a gradient scheme in the table.

7. Moreover, in the text there are some typographic errors (e.g., ‘cellar’ instead of ‘cellular’ at line 342, delete ‘that’ at line 339, ‘sep-ara-tion’ at lines 457-458, m/z at lines 451, 491 should be in italics, some unnecessary spaces appear).

With kind regards,

Reviewer
